# A Polish Version of the Boston Carpal Tunnel Questionnaire (BCTQ-PL) for Use Among Patients with Carpal Tunnel Syndrome Undergoing Physiotherapy: Translation, Cultural Adaptation, and Validation

**DOI:** 10.3390/healthcare13111288

**Published:** 2025-05-29

**Authors:** Sabina Mastej, Agnieszka Bejer, Anita Pacześniak-Jost, Oliver Dörner, Teresa Pop

**Affiliations:** 1Faculty of Health Sciences and Psychology, Collegium Medicum, University of Rzeszów, Rejtana16C, 35-959 Rzeszow, Poland; sabinajurczyk@wp.pl (S.M.); apaczesniak@ur.edu.pl (A.P.-J.); popter@interia.pl (T.P.); 2Physiotherapy Unit, Jasło Health Center, 21 Stanisława Staszica Street, 38-200 Jasło, Poland; 3Krankenhaus Buchholz und Winsen Medizinische Klinik, Friedrich-Lichtenauer-Allee 1, 21423 Winsen (Luhe), Germany; oliver.doerner@krankenhaus-buchholz.de

**Keywords:** carpal tunnel syndrome, BCTQ, validation, psychometric properties, cultural adaptation, extracorporeal shock wave therapy

## Abstract

**Objectives**: The cultural and linguistic adaptation of the Boston Carpal Tunnel Questionnaire (BCTQ) to Polish and the assessment of its psychometric properties among patients undergoing extracorporeal shock wave therapy (ESWT). **Methods**: This was a cross-sectional study with repeated measures during retest examinations. Subjects from an outpatient rehabilitation center in Poland (n = 103) with mild to moderate carpal tunnel syndrome (CTS) were evaluated three times. Test 1 and test 3 (after a series of four treatments using EWST) included the following: completing the BCTQ, QuickDASH, and SF-36 questionnaires, the VAS pain scale, performing the Tinel–Hoffmann and Phalen tests, and an assessment of grip strength. Test 2 (test–retest BCTQ) was performed two to seven days after test 1. **Results**: The Polish version of the BCTQ demonstrated a high internal consistency, with a Cronbach’s alpha of 0.861 for the Symptom Severity Scale (SSS) and 0.924 for the Functional Status Scale (FSS). It also showed excellent test–retest reliability, with Intraclass Correlation Coefficients (ICCs) of 0.941 for the SSS and 0.925 for the FSS. The Standard Error of Measurement (SEM) was 0.16 for the SSS and 0.21 for the FSS, while the Minimal Detectable Change (MDC) was 0.43 and 0.59, respectively. It has a high construct validity as 80% of the a priori adopted hypotheses were confirmed. The mean decrease after ESWT on the SSS was 1.04 points, and on the FSS was 0.77 points. The ES value for the SSS scale was 1.62 and for FSS 0.99, and the SRM was 1.35 for the SSS, and 1.01 for the FSS, which proves a higher sensitivity to changes in the BCTQ-PL. **Conclusions**: The BCTQ-PL is a valid and reliable tool for assessing CTS-related symptoms and functional status in Polish-speaking patients.

## 1. Introduction

Carpal tunnel syndrome (CTS) is the most common peripheral neuropathy of the upper limb. It is characterized by pain and paresthesia in the area supplied by the median nerve [1]. The American Academy of Orthopedic Surgeons (AAOS) and the American Society for Surgery of the Hand (ASSH) provide guidelines necessary for the diagnosis of CTS.

In addition to a clinical examination and specific imaging tests, they also include the use of standardized Patient-Reported Outcome Measures (PROMs), which provide valuable information about the symptoms, their impact on daily functioning, and the effectiveness of selected therapeutic procedures from the patient’s perspective [2]. There are over a dozen English-language-specific PROMs for the assessment of the upper limb. Some of them are used for the subjective assessment of patients with dysfunction of the entire upper limb, such as the Disabilities Arm, Shoulder, and Hand Questionnaire (DASH) and its shortened version, the QuickDASH [3], the American Shoulder and Elbow Surgeons questionnaire (ASES) [4], or the Shoulder Pain and Disabilities Index questionnaire (SPADI) [5]. Others, such as the Michigan Hand Outcomes Questionnaire (MHQ), are specific to the area of the hand [6]. All of them are characterized by good psychometric properties, but they do not contain items directly related to CTS, offering only general information on the degree of dysfunction or the severity of symptoms in the upper limb.

In the case of CTS, the most frequently used questionnaire is the Boston Carpal Tunnel Questionnaire (BCTQ) [7], developed by Levine et al. in 1993, which is used in many different countries. Available language versions include Chinese [8], Japanese [9], Korean [10], Portuguese [11], Spanish [12], Thai [13], and Turkish [14,15], among others [16,17]. The results of studies using the BCTQ are easily and measurably comparable. In 2006, a systematic review of the psychometric properties of the BCTQ was published by Leite et al., which confirms that it is a standardized tool for determining the severity of symptoms and the functional status of patients with CTS. It is characterized by good psychometric properties, such as validity, reliability, and sensitivity to clinical changes [18].

The aim of our study was to conduct the cultural and linguistic adaptation of the BCTQ to a Polish version and to assess its psychometric properties—reliability, validity, and responsiveness—among patients with CTS undergoing physiotherapy.

## 2. Materials and Methods

### 2.1. Participants

Between April and August 2019, we initially recruited 126 patients diagnosed with CTS for the study. Patients were selected based on an interview with either an orthopedist or neurologist and had an ultrasound (n = 50) or electromyography (EMG; n = 53). Ultimately, 103 patients diagnosed with mild or moderate CTS were chosen for the study. We took into account factors that would exclude patients from extracorporeal shock wave therapy, such as implantable cardioverter defibrillator, cardiovascular failure, acute inflammation, fever, blood coagulation disorders and the use of anticoagulants, pregnancy, neoplastic disease, previous injuries or fractures of the upper limb, congenital defects of the upper limb, neurological diseases (e.g., Parkinson’s disease, multiple sclerosis, amyotrophic lateral sclerosis, polyneuropathy, and stroke), advanced degenerative and rheumatic processes involving the upper limb, cervical discopathy at the C5-Th1 with root compression, taking medications that worsen psychophysical fitness, and inability to complete the questionnaires (aphasia and dementia disorders). All patients were native Polish speakers. This study was conducted at the outpatient physiotherapy unit of Jasło Health Center, Poland. Figure 1 presents a flow chart depicting the recruitment process and subsequent testing stages of the patients.

### 2.2. Sample Size

A post-hoc analysis of the test power significance was conducted for Intraclass Correlation Coefficients (ICCs) for the null hypothesis ICC = 0.7 with a sample size of 103 individuals, significance level of 0.05, and the expected ICC value in our population. The test power is extremely high with over 0.999 for each scale. It showed that the sample size was statistically satisfactory.

### 2.3. Design

This was a cross-sectional study with repeated measures during retest examinations. Before conducting the research, on 31 October 2017, consent was obtained from the authors of the source version of the BCTQ, represented by Professor Barry Simmons, for the translation and cultural adaptation of the BCTQ to Polish and for the assessment of its psychometric properties.

Stage 1 involved translation and cultural adaptation of the BCTQ into Polish.

The process of translation and language adaptation was conducted in line with the international guidelines proposed by Beaton et al. [19]. Figure 2 presents the six steps of the linguistic and cultural adaptation of the BCTQ-PL.

Stage 2 involved a prospective evaluation of the essential psychometric properties of the BCTQ-PL.

The patients (n = 103) underwent three tests. During test I the patients completed the Polish versions of all the questionnaires. During test II, 2–7 days after test I (retest), the patients completed only the BCTQ. After completing the questionnaire, each patient underwent the first extracorporeal shock wave therapy (ESWT): a series of four treatments at weekly intervals, frequency 10 Hz, pressure 2 Ba, number of strokes 2500, focus transmitter 15 mm, and average energy density 0.22 mJ/mm^2^. Test III was performed 3 months after the completion of the full series of ESWT and involved filling in the Polish versions of all the questionnaires. All the patients answered all the questions.

### 2.4. Measurements

#### 2.4.1. PROMs

##### Boston Carpal Tunnel Questionnaire (BCTQ)—Polish Version (PL)

The questionnaire consists of two scales. The Symptom Severity Scale (SSS) has 11 items, including the degree of pain experienced by the patient during day and night and the frequency of painful episodes including numbness, and weakness and tingling in the hand, as well as difficulty grasping and handling small objects. The Functional Status Scale (FSS), on the other hand, includes eight items related to functionality, including difficulty in carrying out such activities as writing, buttoning, holding a book, using a telephone, opening jars, housework, carrying shopping bags, washing, and dressing. The responses are given on a 5-point scale, where 1 means the lowest degree of symptoms/no difficulties with a given activity, and 5 means the most severe symptoms/inability to perform a given activity. Each scale produces a final score in the range of 1–5, which is the total of all the responses in the questionnaire divided by the number of items answered. The result is rounded to 1/100. The higher the score on the five-point scale, the greater the degree of hand disability [7].

##### Disabilities of the Arm, Shoulder, and Hand Questionnaire (QuickDASH)—Shortened Polish Version

The questionnaire contains 11 items related to the difficulties in the performance of activities using the upper limb, as well as the symptoms and their impact on social activities, work, and sleep. The score is in the range of 0–100, with the higher score corresponding to a greater impediment in the upper limb [3].

##### The Short-Form 36 Health Survey (SF-36 Version 2.0)—Polish Version

The questionnaire includes eight domains: Physical Functioning, Role Limitations due to physical health, Bodily Pain, General Health perceptions, Vitality, Social Functioning, Role Limitations due to emotional problems, and Mental Health. In addition, the first four domains constitute the Physical Component Scale (PCS), whereas the next four constitute the Mental Component Scale (MCS). The results are in the range of 0–100 points, with a score of 0 corresponding to the worst and a score of 100 reflecting the best quality of life [20].

##### Visual Analogue Scale (VAS)

The scale is used to determine the intensity of pain experienced by the patient. A point corresponding to the patient’s pain is marked on a 10 cm scale, where 0 corresponds to no pain and 10 reflects the worst pain imaginable [21].

#### 2.4.2. Provocative Tests

##### Phalen’s Test

During the test. the patient performs palmar flexion of the wrist for 60–120 s. A positive test result is recorded (representing a median nerve injury) if, after this time, the patient reports increased symptoms, e.g., tingling and numbness [22,23].

##### Hoffman–Tinel Sign

This test is performed by gently tapping, with the index finger, on the median nerve trunk at the level of the carpal flexion crease. The test result is considered positive (representing a median nerve injury) if radiating pain and paresthesia occur in the area supplied by the median nerve [22,23].

#### 2.4.3. Muscle Strength Test

##### Hand Grip Strength Test with Dynamometer

This was performed using a Jamar® Hand dynamometer [kg] (Fabrication Enterprises Inc., White Plains, NY, USA; manufactured in China), in compliance with the guidelines of the American Society for Surgery of the Hand (ASSH) [24,25]. Three measurements were carried out and the arithmetic mean was calculated from these.

### 2.5. Statistical Analyses

The authors used SPSS Statistics software version 24. It was assumed a level of statistical significance was reflected by *p* < 0.05. The distribution of the results was verified using the Kolmogorov–Smirnov test.

#### 2.5.1. Reliability

Internal consistency was assessed using the Cronbach’s coefficient (which should be in the range between 0.70 and 0.95) [26].Repeatability was assessed by comparing the baseline test performed using the BCTQ-PL questionnaire (test) with the test repeated after 2–7 days (retest). The following tools were applied for this purpose:

Intraclass correlation coefficient (ICC2.1) with values lower than 0.5, between 0.5–0.75, between 0.75–0.9, and over 0.9 indicating poor, moderate, good, and excellent reliability, respectively [27].Spearman’s rank correlation coefficient (SCC) between the two BCTQ-PL measurements. The following correlation strength scale was adopted: |R| < 0.3—lack of correlation; 0.3 ≤ |R| < 0.5—weak correlation; 0.5 ≤ |R| < 0.7—moderate correlation; 0.7 ≤ |R| < 0.9—strong correlation; 0.9 ≤ |R| < 1—very strong correlation; and |R| = 1—perfect correlation.

Measurement error:

Standard error of measurement (SEM) determines to what extent the values of a given measure will differ in subsequent measurements made under the same conditions for purely random reasons [28].Minimal detectable change (MDC) defines the smallest difference between two measurements that (with 95% confidence) does not result only from random fluctuations [29].

#### 2.5.2. Validity

Construct validity

The construct validity was assessed by correlating (SCC) the scores of the BCTQ-PL and the reference questionnaires and tests. The authors evaluated the significance of the relationship between the values obtained in the BCTQ-PL and the results of Hoffman–Tinel sign and Phalen’s test using the *t*-test for independent samples.

The following a priori hypotheses (10) were made: (1) BCTQ-PL SSS will strongly correlate with QuickDASH (both tools assess the symptom severity and the functional impact of upper limb conditions, due to which a strong correlation is theoretically expected); (2) BCTQ-PL FSS will strongly correlate with QuickDASH (FSS and QuickDASH both focus on functional limitations of the upper extremity, the fact being conducive to a strong correlation); (3) BCTQ-PL SSS will strongly correlate with VAS (SSS covers pain and sensory disturbances, which are core factors measured by the VAS); (4) BCTQ-PL FSS will correlate moderately or weakly with VAS (FSS measures functional ability rather than pain intensity, which suggests a weaker theoretical link with VAS); (5) correlations between BCTQ-PL and QuickDASH will be stronger than those between BCTQ-PL and SF-36 (QuickDASH is disease-specific for the upper limb function, whereas SF-36 is a generic quality-of-life tool, leading to expected stronger correlations between the former and the BCTQ-PL); (6) correlations between FSS and SF-36 will be stronger than between SSS and SF-36 (FSS relates more directly to daily functioning, which aligns better with the functional domains assessed by SF-36); (7) correlations between FSS and SF-36 PCS will be stronger than between FSS and SF-36 MCS (FSS reflects physical disability, thus aligning more closely with the PCS of SF-36); (8) correlations between SSS and SF-36 PCS will be stronger than between SSS and SF-36 MCS (symptom severity in CTS, e.g., pain and numbness, impacts physical health more directly than mental health); (9) both BCTQ-PL scales will correlate moderately or weakly with hand grip strength (grip strength is an objective functional test, but the function is often preserved in mild-to-moderate CTS, leading to limited correlation with subjective symptom reports); and (10) both BCTQ-PL scales will significantly differentiate patients with positive vs. negative results on Tinel–Hoffmann and Phalen’s tests (these provocative tests are diagnostic tools for CTS, so differences in SSS and FSS scores between positive and negative groups are theoretically expected). According to the criteria proposed by Terwee et al. [26], construct validity of the BCTQ-PL is deemed acceptable when at least 75% of a priori hypotheses are confirmed in a sample of no fewer than 50 participants.

Structural validity

The unidimensionality of the BCTQ-PL questionnaire was analyzed separately for both scales. For this purpose, exploratory factor analysis (EFA) was applied to examine the number of dimensions of the SSS and FSS, and to reduce the information contained in the original detailed questions to a smaller number of directly unobservable factors (an eigenvalue greater than 1.0, as well as an explained variance of more than 10% were assumed for each subscale). The Kaiser–Meyer–Olkin measure of the sampling adequacy (KMO) was set at >0.70 to indicate adequate sampling, and the significance level of the Barlett Test of Sphericity was *p* < 0.001, indicating that the EFA could be used for data analysis.

#### 2.5.3. Responsiveness

The Wilcoxon test was used to assess the significance of the changes in the BCTQ-PL scores before (test I) and after physiotherapy (test III).Effect size (ES) was calculated to detect clinical changes in patients receiving physiotherapy. Absolute ES values ≤ 0.20 represent low responsiveness, values in the range of 0.21–0.79 reflect moderate responsiveness, and values ≥ 0.80 indicate a high responsiveness of the assessment tool used to detect clinical changes in the patient’s condition.Standardized response mean (SRM) was interpreted as in the case of ES [30,31,32].

### 2.6. Ethical Considerations

The study was approved by the institutional Bioethics Committee at the University of Rzeszow, Poland (Resolution No. 5/01/2019).

## 3. Results

### 3.1. Stage 1—BCTQ Translation and Cultural Adaptation

The translation and cross-cultural adaptation of the BCTQ to Polish required several changes to be introduced during the process (Appendix A):

Step 1. Preliminary translation and Step 2. Translation synthesis.

Most of the discrepancies in the two versions of the translation (T1 and T2) were related to only small differences in specific expressions, vocabulary, or word order.

Step 3. Back translation.

The necessity to adjust item 4 of the FSS “Gripping of a telephone handle” to technological progress was identified and reported to the expert committee. Furthermore, a discrepancy in the translations of item number 8 of the FSS “Bathing and dressing” was noticed.

Step 4. Review by the committee of experts.

Item 4 of the FSS “Gripping of a telephone handle” was updated to “Grabbing and using a mobile”. Item 8 of the FSS, “Bathing and dressing”, was modified to ensure semantic equivalence in Polish. In the remaining questions, there were only minor discrepancies, exclusively in the grammatical forms.

Step 5. Test of the pre-final version.

The average time required by the respondents to complete the questionnaire was 4 min 5 s. Following a review of all the patients’ remarks and comments, minor modifications were made in two items in the SSS (items 2 and 10), on one response option.

Step 6. Final report.

The Polish version of the Boston Carpal Tunnel Syndrome Questionnaire (BCTQ-PL) was approved (Appendix A).

### 3.2. Stage 2—Psychometric Investigation

#### 3.2.1. The Clinical Characteristics of the Patients

The total sample enrolled for the study included 103 patients, aged = 55.7 ± 11.2 years, in the range of 25–85 years, and females (n = 78) accounted for 75.7% of the group. The characteristics of the problems, the affected area, history of the treatment, and comorbidities are presented in Table 1.

The values presented in Table 2 and Table 3 are related to the PROMs and muscle strength test.

#### 3.2.2. Internal Consistency

As shown in Table 4, on average, the severity of the symptoms decreased by 1.04, and the functional status improved by 0.77. The values of ES and SRM were, respectively, 1.62 and 1.35 for the SSS, and 0.99 and 1.01 for the FSS. According to the adopted interval criterion (ES > 0.8; SRM > 0.8), the results indicate a high responsiveness of both the SSS and FSS to clinical changes after the physiotherapy applied in the study. The values of the Cronbach’s coefficient for both the SSS (0.861) and the FSS (0.924) indicate a very good internal consistency of the scales. All partial Cronbach’s α coefficients (calculated when removing individual items from the questionnaire) are in the range of 0.842–0.918; therefore, no item of the questionnaire negatively affects the internal consistency of both scales (Table 4).

#### 3.2.3. Test–Retest Reliability and Measurement Error

The ICC2.1 value for both scales is at a high level—above 0.90. The values of measurement errors, SEM and MDC95%CI, are shown in Table 5. In addition, the correlations (SCC) between the two BCTQ-PL measurements were also high, r = 0.93 (*p* < 0.0001) for SSS, and r = 0.90 (*p* < 0.0001) for FSS, which further indicates good test–retest reliability.

#### 3.2.4. Construct Validity

As shown in Table 6 and Table 7, the a priori hypotheses No. 1 through 8 were confirmed, while hypotheses No. 9 and 10 were rejected. The construct validity of the BCTQ-PL received a positive rating according to the criteria proposed by Terwee et al. [26], as 80% of the results were consistent with the predefined hypotheses.

#### 3.2.5. Structural Validity Analysis

The KMO measure of the sampling adequacy was 0.792 for the SSS and 0.909 for the FSS, indicating a good fit for factor analysis. Bartlett’s test of sphericity was significant for both the SSS and FSS (*p* < 0.001), suggesting that the correlation matrices differ significantly from the identity matrix and justifying the use of the EFA.

The results of the EFA for the BCTQ-PL are presented in Table 8. The information contained in the 11 items on the SSS could be reduced to two factors. The first factor contained 39.6% of the information enclosed in the 11th item pattern, while the second factor contained 27.0%. After reducing the information contained in the original 11 items to two factors, it was possible to recreate almost two-thirds (approx. 66%) of the initial variability. The first factor included items no. 1, 2, 6, 8, 9, and 10 (night symptoms, not directly related to activities of daily living), while the second factor included items no. 3, 4, 5, 7, and 11 (daytime symptoms related to functionality). In the EFA for the FSS, a single factor was identified, which correlated with all component items of the scale (ranging from 0.78 to 0.84) and made it possible to explain almost two-thirds (65.7%) of the information contained in the initial data, which comprised answers to eight component questions.

#### 3.2.6. Responsiveness

For both BCTQ-PL scales, very clear, positive changes were observed three months after the applied shock wave therapy (*p* < 0.001). As shown in Table 9, the mean decrease in the severity of symptoms was 1.04, and the functional status improved by 0.77. The value of ES and SRM were, respectively, 1.62 and 1.35 for the SSS, and 0.99 and 1.01 for the FSS. According to the adopted interval criterion (ES > 0.8; SRM > 0.8), the results indicate a high responsiveness of the SSS and FSS to clinical changes after the physiotherapy applied in the study.

## 4. Discussion

This paper presents the process of translation and cultural adaptation of the BCTQ to a Polish version, which was carried out in accordance with international guidelines [14,18,19]. During this stage of the study, no problems were identified and a conceptual equivalence of the BCTQ-PL with the source version was obtained. The subsequent validation analyses demonstrated that the Polish version of BCTQ is characterized by high reliability, validity, and responsiveness to clinical changes, which is in line with the findings reported by the authors of other-language versions [33,34,35].

The reliability of the questionnaire was investigated along with an analysis of the internal consistency and repeatability of BCTQ-PL. Internal consistency was determined by calculating the Cronbach’s α coefficient, which was ≥0.70, as anticipated. Our results are almost identical to those obtained for the original version of the questionnaire, with values of 0.89 for the BCTQ SSS and 0.91 for the FSS [7]. Similarly, high values of Cronbach’s α were identified in the study by Kim et al., i.e., 0.89 for the SSS and 0.90 for FSS [36], and by other researchers [9,10,16,17,34].

In order to assess the repeatability of the BCTQ-PL questionnaire, a two-to-seven-day interval between the test and re-test was used. This interval was long enough for the patients to forget the questions from test 1, but short enough so that their state of health did not change [9,10,15,33]. The good repeatability of the questionnaire was shown by the very high ICC values (over 0.90 for both scales), and the finding was consistent with the results reported in other studies [8,9,10,15,16,36,37]. Only the Persian version was found to present a poorer repeatability than the other language versions, as reflected by the ICC values of 0.53 for the SSS and 0.77 for the FSS [17]. In the source version, the repeatability of the tool was measured by performing a test–retest procedure on two consecutive days, and using the Pearson’s coefficient. The result was 0.91 for the SSS and 0.93 for the FSS, which also shows excellent repeatability [7]. The test–retest analysis also included the calculation of the SEM and the MDC. The current findings related to the MDC indicate that clinicians and researchers may consider that the differences in the SSS at the level of at least 0.43, and in the FSS at the level of at least 0.59 are dependable. Higher MDC values were reported by the authors of the Chinese version of the questionnaire, i.e., 0.86 for SSS and 0.75 for FSS [8]. However, in the Arabic version of the questionnaire validated by Alanazy et al., the MDC amounted to 4.7 and 4.5 for the SSS and the FSS, respectively [37].

As a result of the analyses concerning the construct validity assessment, eight out of ten (i.e., 80%) a priori hypotheses were confirmed. In accordance with the guidelines presented by Terwee et al. [26], this result indicates an acceptable construct validity of the BCTQ-PL. Two a priori hypotheses, the ones related to hand grip strength and the results of provocative tests (Phalen’s test and Tinel–Hoffmann sign), were not confirmed. This may be explained by the fact that mild to moderate CTS cases were included in our sample (SSS $\bar{x}$ = 2.98); in such cases, global hand grip strength often remains relatively preserved. Additionally, the sensitivity of provocative tests varies widely in the literature (ranging from 42% to 85% for Phalen’s test and from 38% to 100% for the Hoffmann–Tinel sign [22]), which may explain the limited discriminant power of the BCTQ-PL in differentiating between patients with positive and negative results. These findings further highlight the complexity of CTS evaluation and support the need for a multidimensional assessment approach that combines both subjective and objective measures. The validity of the original version of BCTQ was confirmed using the correlation between the SSS and FSS and such measures as grip strength, pinch strength, the two-point discrimination test, and the Semmes–Weinstein monofilament test, as well as the velocity of midrange sensory conduction [7]. Similar to our study, the correlation was compiled by researchers from Japan [9]. In the Dutch version of the BCTQ, the authors assessed the validity by investigating the correlation between the BCTQ and the Likert scale, grip strength, ultrasound and electrophysiological examinations; however, no correlation between BCTQ and the above measures was found [33].

We examined the unidimensionality of the BCTQ-PL using an EFA separately for both subscales. The FSS presented the unidimensional structure, while the SSS showed a correlation of items in two subgroups. Items no. 1, 2, 6, 8, 9, and 10 addressing nocturnal symptoms were not related to everyday life, and items no. 3, 4, 5, 7, and 11 were more related to daytime symptoms concerning functionality. Very few BCTQ validation studies performed a similar factor analysis as reported in the current study [9,33,38]. The factor analysis of the Dutch version of the BCTQ, in a study in which the patients were examined before and after the surgical intervention, showed similar findings to those reported in the present study as regards the unidimensionality of the FSS. However, the SSS completed by patients before the surgery, according to the latter authors, appear to measure three different subsets of items, namely, “daytime symptoms” (α = 0.80), “night symptoms” (α = 0.83), and “hand manipulation abilities” (α = 0.72). After the surgical intervention, the factor analysis of the SSS changed further and the researchers distinguished only two subgroups of questions. The subsets of items related to “nocturnal symptoms” and “hand manipulation skills” merged into one coherent group of questions, while the “daytime symptoms” items remained as a separate subgroup [33]. Imadea et al., in the Japanese version of BCTQ, also distinguished between two factors in the SSS [9]. These findings indicate the need for further research using such analyses as the CFA or Rasch modeling to confirm the factor structure of the BCTQ-PL.

In the process of assessing the responsiveness of the BCTQ-PL to clinical changes, very clear, positive effects were observed following the physiotherapy applied in the study (*p* < 0.001). The severity of symptoms, on average, decreased by 1.04 and the functional status of the patients improved by about 0.77. The study validating the original version reported a mean decrease in the SSS after surgery by 1.5, and the FSS score improved by 1.0 [7]. In the present study, the ES values were very high (1.62 and 0.99 for the SSS and FSS, respectively), which proves that the BCTQ-PL is highly responsive to clinical changes after therapy. The ES values in the present study were almost identical to those reported for the original version of the BCTQ [7] and consistent with the findings of other researchers [9,33,38]. The SRM values obtained for both scales also confirmed a high responsiveness of the BCTQ-PL. Similar SRM values were reported in the case of the Dutch version (SSS-1.49 and FSS-0.76) [33] and slightly lower values were found in the Chinese version (SSS-1.03 and FSS-0.62) [8].

### Limitations and Future Considerations

The limitations of the present study include the fact that no assessment of psychometric properties was performed on patients undergoing other forms of intervention in CTS, such as the commonly used median nerve decompression surgery, injections of corticosteroids, or other types of physical therapy. Notably, the exclusion criteria applied, some of which were directly related to the contraindications for ESWT, may have inadvertently introduced sample imbalances, particularly with respect to comorbidities and the distribution of CTS in dominant vs. non-dominant limbs. Future studies should aim to validate the BCTQ-PL in heterogeneous groups of patients receiving treatment based on different therapeutic approaches and recruited from various healthcare centers, in order to improve the generalizability of the findings. Although one objective clinical measure (hand grip strength) was included, the validation process primarily relied on subjective PROMs. To further strengthen the construct validity of the BCTQ-PL, future studies should include additional objective clinical assessments, such as electroneurography or electromyography. A key limitation of the present study is that, although the SSS and FSS were used as separate scales in line with the original and other language versions of the BCTQ, the EFA suggested the possibility of a two-factor structure within the SSS. Future research should verify this finding using a confirmatory factor analysis (CFA) or Rasch modeling on larger and more diverse samples to assess whether distinguishing subscales for “nocturnal symptoms” and “daytime symptoms related to functionality“ would improve structural validity.

## 5. Conclusions

The Polish version of the BCTQ, a product of a reliable translation and effective cultural adaptation to Polish, demonstrates high reliability, validity in measuring the severity of symptoms and dysfunctions characteristic of patients with CTS, and responsiveness to clinical changes in patients’ health after the physiotherapy was applied. The BCTQ-PL can successfully be used in Poland both in clinical work and research projects of national and international importance regarding patients with CTS.

## Figures and Tables

**Figure 1 healthcare-13-01288-f001:**
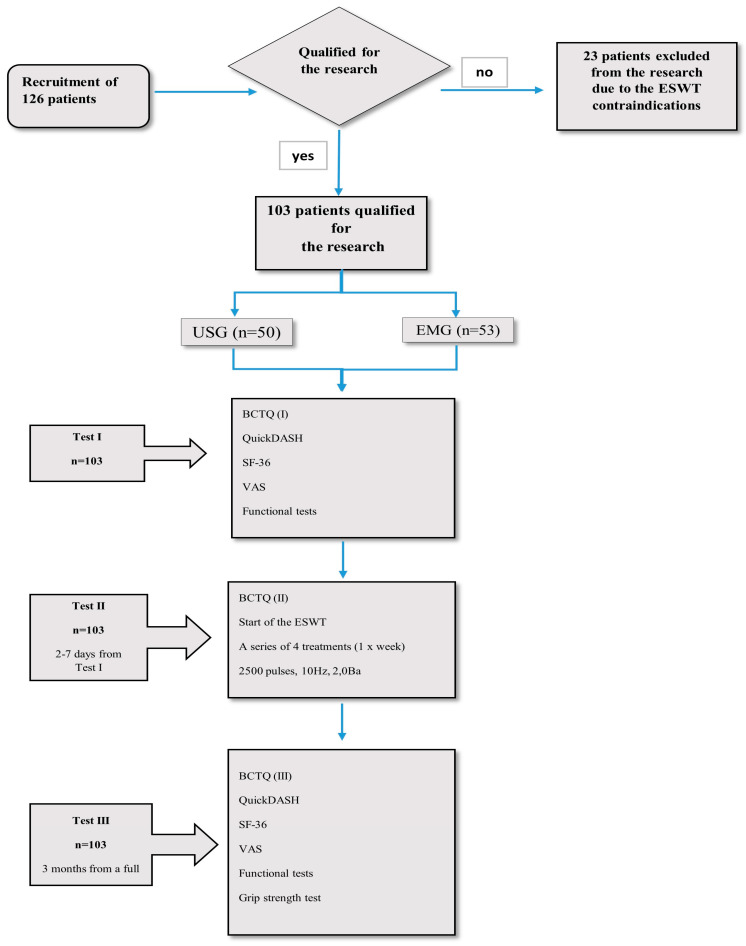
Flow chart of patient recruitment and testing stages.

**Figure 2 healthcare-13-01288-f002:**
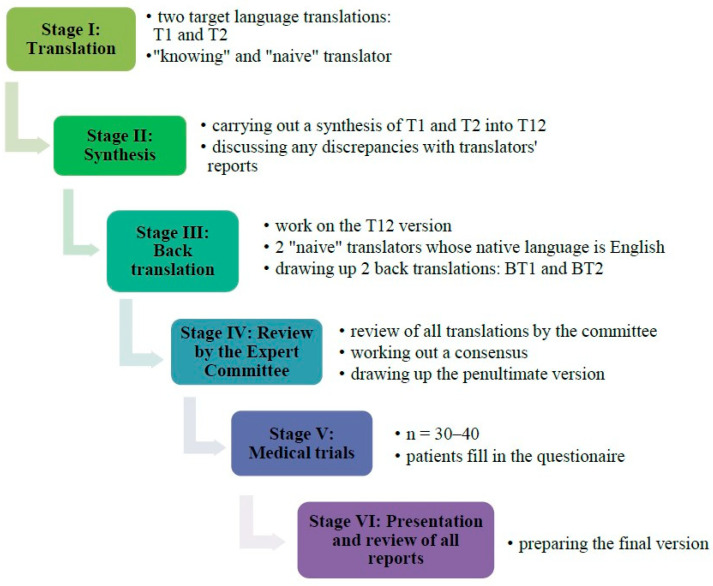
Stages of linguistic and cultural adaptation of the BCTQ-PL.

**Table 1 healthcare-13-01288-t001:** The clinical characteristics of the patients (n = 103).

	n	%
The duration of hand dysfunction		
<1 month	2	1.9%
1–3 months	8	7.8%
4–8 months	16	15.5%
>8 months	77	74.8%
CTS side	n	%
Right	59	57.3%
Left	44	42.7%
Limb affected by CTS	n	%
Dominant	64	62.1%
Non-dominant	39	37.9%
Previous CTS treatment	n	%
No treatment	62	60.2%
Pharmacological treatment	25	24.3%
Physiotherapy	12	11.7%
Orthopedic equipment	4	3.9%
Co-existing diseases	n	% *
Hypertension	30	29.1%
Hyperthyroidism/hypothyroidism	14	13.6%
Diabetes	8	7.8%
Prostatic hypertrophy	5	4.9%
Discopathy of the lumbar spine	3	2.9%
Elevated cholesterol levels	2	1.9%
Bronchial asthma	2	1.9%
Sarcoidosis	1	1.0%
Irritable bowel syndrome	1	1.0%
Pancreatic diseases	1	1.0%
Kidney failure	1	1.0%
Varicose veins of the lower extremities	1	1.0%
No comorbidities	51	49.5%

Abbreviations: n, group size; %, percentage of the study group; CTS, carpal tunnel syndrome. * The total does not have to be 100% as it was possible to report any number of conditions.

**Table 2 healthcare-13-01288-t002:** The absolute values of all PROMs and muscle strength test (n = 103).

Questionnaire	Test I			Test II			Test III		
	$\bar{x}$	Me	s	$\bar{x}$	Me	s	$\bar{x}$	Me	s
BCTQ-PL SSS	2.98	3.00	0.64	2.95	2.91	0.63	1.93	1.91	0.68
BCTQ-PL FSS	2.72	2.88	0.78	2.70	2.88	0.78	1.95	1.88	0.72
QuickDASH	49.1	52.3	17.7				25.9	22.7	18.5
VAS	5.7	6.0	1.7				2.6	2.5	1.9
SF-36—PCS									
Physical functioning	65.1	70.0	21.9				69.4	70.0	22.7
Role physical	54.2	50.0	20.8				59.8	56.3	21.3
Body pain	45.8	45.0	19.9				62.5	67.5	22.8
General health	52.7	50.0	16.7				53.1	50.0	17.3
PCS—Total score	58.2	59.0	17.5				63.0	60.5	18.6
SF-36—MCS									
Vitality	53.2	50.0	16.9				54.9	56.3	16.0
Social functioning	66.9	62.5	22.5				72.7	75.0	20.8
Role emotional	73.1	75.0	23.8				76.8	83.3	23.6
Mental health	61.1	60.0	16.9				64.2	65.0	16.6
MCS—Total score	62.2	60.7	16.6				65.4	66.1	16.1
SF-36—Total score	60.2	59.6	15.9				64.2	64.4	16.2
**Muscle strength test**	**Test I**			**Test II**			**Test III**		
Hand grip strength	21.1	19.7	8.7				28.8	28.3	8.8

Abbreviations: $\bar{x}$, mean; Me, median; s, standard deviation; BCTQ-PL, Boston Carpal Tunnel Questionnaire—Polish version; SSS, Symptom Severity Scale; FSS, Functional Status Scale; SF-36, Short-Form Medical Outcome Study; VAS, Visual Analog Scale; PCS, Physical Component Scale; MCS, Mental Component Scale.

**Table 3 healthcare-13-01288-t003:** The values of Phalen test and Tinel–Hoffmann sign (n = 103).

Test Result	Test I				Test III			
	Phalen Test		Tinel–Hoffmann Sign		Phalen Test		Tinel–Hoffmann Sign	
	n	%	n	%	n	%	n	%
Positive	75	72.8%	73	70.9%	25	24.3	21	20.4
Negative	28	27.2%	30	29.1%	78	75.7	82	79.6

Abbreviations: n, group size; %, percentage of the study group.

**Table 4 healthcare-13-01288-t004:** Internal consistency of the BCTQ-PL (n = 103).

SSS	$\bar{x}$	*s*	Cronbach α	FSS	$\bar{x}$	*s*	Cronbach α
1	3.04	1.02	0.847	1	2.45	0.98	0.918
2	2.99	1.01	0.845	2	2.55	0.97	0.911
3	2.80	0.88	0.848	3	2.40	0.94	0.913
4	3.06	1.20	0.862	4	2.52	0.89	0.914
5	2.81	1.18	0.866	5	3.50	1.08	0.916
6	3.15	0.89	0.846	6	2.94	0.92	0.915
7	2.88	0.90	0.849	7	3.10	1.04	0.912
8	3.07	0.85	0.848	8	2.33	0.86	0.916
9	3.31	0.93	0.846				
10	3.09	0.97	0.847				
11	2.55	0.99	0.842				
Total	2.98	0.64	0.861	Total	2.72	0.78	0.924

Abbreviations: BCTQ-PL, Boston Carpal Tunnel Questionnaire—Polish version; SSS, Symptom Severity Scale; FSS, Functional Status Scale; n, group size; *s*, standard deviation; $\bar{x}$, mean.

**Table 5 healthcare-13-01288-t005:** Test–Retest Reliability and Measurement Error of the BCTQ-PL (n = 103).

BCTQ-PL	ICC2.1 (95%CI)	SEM	MDC (95%CI)
SSS	0.941 (0.914–0.959)	0.16	0.43
FSS	0.925 (0.891–0.949)	0.21	0.59

Abbreviations: BCTQ-PL, Boston Carpal Tunnel Questionnaire—Polish version; SSS, Symptom Severity Scale; FSS, Functional Status Scale; ICC, Intraclass Correlation Coefficient; SEM, Standard Error of Measurement; MDC, Minimal Detectable Change.

**Table 6 healthcare-13-01288-t006:** Correlations between the BCTQ-PL and reference questionnaires as well as muscle strength test.

Questionnaire	BCTQ-PL (n = 103)	
	SSS	FSS
QuickDASH Total	r = 0.70, *p* < 0.001 ***	r = 0.80, *p* < 0.001 ***
VAS Pain	r = 0.73, *p* < 0.001 ***	r = 0.47, *p* < 0.001 ***
SF-36—PCS		
Physical functioning	−0.38, *p* = 0.000 ***	−0.56, *p* < 0.001 ***
Role physical	−0.38, *p* = 0.000 ***	−0.54, *p* < 0.001 ***
Body pain	−0.52, *p* < 0.001 ***	−0.56, *p* < 0.001 ***
General health	−0.10, *p* = 0.312	−0.30, *p* = 0.002 **
PCS—Total score	−0.38, *p* = 0.000 ***	−0.57, *p* < 0.001 ***
SF-36—MCS		
Vitality	−0.24, *p* = 0.016 *	−0.33, *p* = 0.001 ***
Social functioning	−0.28, *p* = 0.004 **	−0.38, *p* = 0.000 ***
Role emotional	−0.14, *p* = 0.165	−0.31, *p* = 0.001 **
Mental health	−0.23, *p* = 0.021 *	−0.32, *p* = 0.001 ***
MCS—Total score	−0.25, *p* = 0.010 **	−0.40, *p* < 0.001 ***
SF-36 Total	−0.32, *p* = 0.001 ***	−0.50, *p* < 0.001 ***
**Muscle strength test**	**SSS**	**FSS**
Hand grip strength	r = −0.20, *p* = 0.043	r = −0.21, *p* = 0.033 *

Abbreviations: BCTQ-PL, Boston Carpal Tunnel Questionnaire—Polish version; SSS, Symptom Severity Scale; FSS, Functional Status Scale; QuickDASH, Disabilities Arm, Shoulder, and Hand Questionnaire; VAS; Visual Analog Scale; SF–36, Short-Form Medical Outcome Study; PCS, Physical Component Scale; MCS, Mental Component Scale. Statistically significant * *p* < 0.05; ** *p* < 0.01; *** *p* < 0.001.

**Table 7 healthcare-13-01288-t007:** Relationship between the BCTQ-PL and the Phalen test, as well as the Tinel–Hoffmann sign.

BCTQ-PL	Phalen Test										*p*
	Positive (n = 75)					Negative (n = 28)					
	$\bar{x}$	Me	*s*	Min	Max	$\bar{x}$	Me	*s*	Min	Max	
SSS	3.03	3.00	0.60	1.64	4.45	2.83	2.95	0.73	1.27	4.36	0.162
FSS	2.83	2.88	0.72	1.00	4.38	2.43	2.50	0.85	1.00	4.00	0.020 *
**Tinel–Hoffmann Sign**											
SSS	3.02	3.09	0.62	1.55	4.45	2.87	2.95	0.69	1.27	4.36	0.296
FSS	2.72	2.88	0.79	1.00	4.38	2.75	3.00	0.75	1.38	4.00	0.860

Abbreviations: BCTQ-PL, Boston Carpal Tunnel Questionnaire—Polish version; SSS, Symptom Severity Scale; FSS, Functional Status Scale; $\bar{x}$, mean; Me, median; *s*, standard deviation; min, minimum values; max, maximum values. *p*, test probability value calculated using the *t*-test for independent samples; * statistically significant (*p* ≤ 0.05).

**Table 8 healthcare-13-01288-t008:** Results of the EFA for the BCTQ-PL (n = 103).

SSS Scale Items	Factor Loadings		FSS Scale Items	Factor Loadings
	Factor 1	Factor 2		Factor 1
SSS-1	0.78	0.09	FSS-1	0.78
SSS-2	0.78	0.12	FSS-2	0.84
SSS-3	0.23	0.80	FSS-3	0.82
SSS-4	−0.00	0.87	FSS-4	0.82
SSS-5	−0.06	0.86	FSS-5	0.80
SSS-6	0.82	0.08	FSS-6	0.80
SSS-7	0.40	0.58	FSS-7	0.83
SSS-8	0.83	0.04	FSS-8	0.80
SSS-9	0.86	0.00		
SSS-10	0.80	0.09		
SSS-11	0.43	0.68		
Eigenvalues	4.36	2.97	Eigenvalues	5.26
Variance Explained	39.6%	27.0%	Variance Explained	65.7%

Abbreviations: BCTQ-PL, Boston Carpal Tunnel Questionnaire—Polish version; SSS, Symptom Severity Scale; FSS, Functional Status Scale; n, group size.

**Table 9 healthcare-13-01288-t009:** Results of the responsiveness analysis (n = 103).

Scale	Test 1	Test 3	Results of Physiotherapy		
	$\bar{x}$ ± *s*	$\bar{x}$ ± *s*	*p*	*ES*	*SRM*
SSS	2.98 ± 0.64	1.93 ± 0.68	*p* < 0.001	1.62	1.35
FSS	2.72 ± 0.78	1.95 ± 0.72	*p* < 0.001	0.99	1.01

Abbreviations: $\bar{x}$, mean; *s* standard deviation; *ES*, effect size; *SRM*, standardized response mean. *p*, test probability value calculated using the *t*-test for independent samples.

## Data Availability

The datasets used and/or analyzed during the current study are available from the corresponding author upon reasonable request.

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
