# Peer review of "A Polish Version of the Boston Carpal Tunnel Questionnaire (BCTQ-PL) for Use Among Patients with Carpal Tunnel Syndrome Undergoing Physiotherapy: Translation, Cultural Adaptation, and Validation"

_healthcare, 2025, doi:10.3390/healthcare13111288_

Round 1
Reviewer 1 Report
Comments and Suggestions for Authors
The authors perform cultural and lingiustic adaptation of BCTQ Polish version and tested its psychometric properties.
The methods section is well written. Authors highlighted adequate exclusion criteria. Additionally, sample size was properly calculated. Study desing is adequate with elaborated each stage during the process of translation and linguistic adaptation. Aside BCTQ authors additionally included Polish versions of QuickDASH and SF-36 version 2.0 as well as VAS, Phalens test, Hoffman-Tinel test and hand grip strength test with dynamometer.
Statistical analysis was described in detail with particular attention to the Reliablity procedure, Validity (construct and structural) procedure and Responsiveness.
Regarding the first subtitle in Results section 3.1. I suggest to move that to methodology and if possible create a table with changes in questionnaire as supplementary file with a table of final Polish version of BCTQ.
Correct commas to dots in tables for percents in Results section.
Please, in results section correct the p values to 3 numbers after the dot so instead of 0.1620 place 0.162 and instead p=0.0000 place p<0.001. Make the changes as well in discussion section.
Limitations are adequatelly addressed.
Conclusions reflect postulated aims.
Literature is representative.
Author Response
Dear Reviewer,
We would like to express our gratitude to you for reviewing our article titled A Polish version of the Boston Carpal Tunnel Questionnaire (BCTQ-PL) for use among patients with Carpal Tunnel Syndrome undergoing physiotherapy: Translation, Cultural Adaptation, and Validation.
Please find detailed responses below and the corresponding corrections in the re-submitted files. The authors perform cultural and linguistic adaptation of BCTQ Polish version and tested its psychometric properties. The methods section is well written. Authors highlighted adequate exclusion criteria. Additionally, sample size was properly calculated. Study desing is adequate with elaborated each stage during the process of translation and linguistic adaptation. Aside BCTQ authors additionally included Polish versions of QuickDASH and SF-36 version 2.0 as well as VAS, Phalens test, Hoffman-Tinel test and hand grip strength test with dynamometer. Statistical analysis was described in detail with particular attention to the Reliablity procedure, Validity (construct and structural) procedure and Responsiveness.
Regarding the first subtitle in Results section 3.1. I suggest to move that to methodology and if possible create a table with changes in questionnaire as supplementary file with a table of final Polish version of BCTQ.
- Thank you for your valuable suggestions. We have created two tables, which are included as supplementary material (Table S1 and Table S2). Table S1 presents the outcomes of the translation and cross-cultural adaptation steps of the BCTQ to the Polish context, while Table S2 contains the Polish version of the BCTQ-PL questionnaire (in Polish). However, we decided to retain Section 3.1. "Stage 1 – BCTQ Translation and Cultural Adaptation" in the Results, as in the Methods section we briefly present the methodology for this process in a flowchart (Figure 2), and in the Results section we summarize the outcomes of the translation and adaptation process, which are further detailed in Table S1.
Correct commas to dots in tables for percents in Results section.
- Thank you for pointing this out. The changes have been made in Tables 1, 3, 6 and 7.
Please, in results section correct the p values to 3 numbers after the dot so instead of 0.1620 place 0.162 and instead p=0.0000 place p<0.001. Make the changes as well in discussion section.
- Thank you for pointing this out. The changes have been made in Tables 5, 6, 7 and 9, as well as in results and discussion sections.
Limitations are adequatelly addressed.
Conclusions reflect postulated aims.
Literature is representative.
Reviewer 2 Report
Comments and Suggestions for Authors
The study addresses a relevant topic; however, I would like to highlight several critical issues that should be addressed to strengthen the manuscript:
Confusion Between Study Objectives
The manuscript simultaneously addresses the validation of the BCTQ-PL and the evaluation of the effects of extracorporeal shock wave therapy (ESWT). This dual focus creates confusion regarding the primary aim of the study. Validation of a patient-reported outcome measure should ideally be conducted independently from treatment effectiveness evaluations to ensure methodological clarity.
Sample Selection Limitation
The study exclusively enrolled patients undergoing ESWT, which may limit the generalizability of the validated questionnaire. For robust validation, a broader CTS patient population, including those undergoing various treatments (e.g., surgical decompression, conservative therapy), should be considered.
Construct Validity Weaknesses
Two of the ten a priori hypotheses were not confirmed, specifically regarding the relationship between BCTQ-PL scores and hand grip strength and clinical tests (Phalen and Tinel-Hoffmann). Although mentioned, this important finding is not critically discussed. The authors should provide a detailed interpretation of these inconsistencies and their implications for the construct validity of the BCTQ-PL.
Issues with Structural Validity
Exploratory factor analysis (EFA) revealed that the Symptom Severity Scale (SSS) is composed of two factors rather than being unidimensional. Despite this, the authors continue to treat it as a single scale without proposing or discussing a potential subscale division. This raises concerns about the structural validity of the SSS, which should be explicitly addressed.
Errors in SF-36 Data Reporting
Anomalous values were found in the SF-36 results, particularly for the Mental Health and Social Functioning domains (e.g., means of 1.7 and 5.8), which are implausible. These inconsistencies suggest potential data reporting errors that should be corrected.
Limited Scope of Validation Methods
The validation process relied solely on subjective patient-reported measures. No comparison with objective clinical evaluations (e.g., electromyography or clinical examination scores) was performed. Including such assessments would considerably strengthen the validation results.
Redundancy and Writing Style
The manuscript contains redundant statements, particularly regarding the reliability and validity of the BCTQ-PL, without adding new information. Improving the conciseness and focusing the discussion on critical methodological issues would enhance the quality of the manuscript.
Potential Conflict of Interest Not Fully Addressed
While no conflicts of interest are declared, validating the questionnaire exclusively in patients treated with ESWT may introduce implicit bias. This aspect should be acknowledged and discussed.
Author Response
Dear Reviewer,
We would like to express our gratitude to you for reviewing our article titled A Polish version of the Boston Carpal Tunnel Questionnaire (BCTQ-PL) for use among patients with Carpal Tunnel Syndrome undergoing physiotherapy: Translation, Cultural Adaptation, and Validation.
Please find detailed responses below and the corresponding corrections highlighted in the re-submitted files.
Comments and Suggestions for Authors
The study addresses a relevant topic; however, I would like to highlight several critical issues that should be addressed to strengthen the manuscript:
Confusion Between Study Objectives
The manuscript simultaneously addresses the validation of the BCTQ-PL and the evaluation of the effects of extracorporeal shock wave therapy (ESWT). This dual focus creates confusion regarding the primary aim of the study. Validation of a patient-reported outcome measure should ideally be conducted independently from treatment effectiveness evaluations to ensure methodological clarity.
Thank you for this important observation. We would like to clarify that the primary objective of our study was the validation of the BCTQ-PL. Due to the clinical context in which patients were referred by physicians for ESWT, we had access to a broad and well-characterized patient group undergoing this specific intervention at the initial stage of treatment. We acknowledge that combining validation and treatment evaluation may lead to methodological ambiguity. For this reason, we have clearly addressed this as a limitation in the 'Limitations and Future Considerations' section (Lines 375-381) and emphasized the need for future research involving more diverse patient populations and other forms of treatment. It is also important to highlight that the validation procedures (e.g., construct validity, reliability, responsiveness) were conducted in accordance with established methodological standards for PROMs and were not dependent on the treatment outcome per se. The observed treatment effect served primarily to assess responsiveness—a key psychometric property—rather than to evaluate the efficacy of ESWT.
Sample Selection Limitation
The study exclusively enrolled patients undergoing ESWT, which may limit the generalizability of the validated questionnaire. For robust validation, a broader CTS patient population, including those undergoing various treatments (e.g., surgical decompression, conservative therapy), should be considered.
Thank you for pointing out this limitation. We agree that enrolling only patients undergoing ESWT constitutes a limitation of our study. For this reason, we have explicitly addressed it in the Limitations and Future Considerations section - Lines 375-381
Construct Validity Weaknesses
Two of the ten a priori hypotheses were not confirmed, specifically regarding the relationship between BCTQ-PL scores and hand grip strength and clinical tests (Phalen and Tinel-Hoffmann). Although mentioned, this important finding is not critically discussed. The authors should provide a detailed interpretation of these inconsistencies and their implications for the construct validity of the BCTQ-PL.
Thank you for this important comment. Indeed, two of the ten a priori hypotheses were not confirmed — specifically those concerning the relationships between the BCTQ-PL scores and hand grip strength, as well as the results of provocative tests (Phalen’s and Tinel-Hoffmann’s signs). Regarding grip strength, it is important to note that in patients with carpal tunnel syndrome (CTS), overall hand grip strength is not always significantly reduced in cases with mild or moderate severity (although prolonged compression of the median nerve may lead to weakness of the thenar muscles and finger flexors, excluding the two heads of the flexor digitorum profundus for digits IV and V). Our patients were qualified for conservative treatment by an orthopedic specialist, which resulted in the inclusion of cases of carpal tunnel syndrome with mild to moderate severity (the mean BCTQ-PL SSS score was 2.98 on a scale from 1 – minimal symptoms to 5 – maximum symptoms; Me = 3.00; SD = 0.64). This may partially explain why hand grip strength did not correlate with the BCTQ-PL scores, which reflect the subjective severity of various symptoms and the resulting functional limitations. Regarding provocative tests such as Phalen’s test and the Hoffmann-Tinel sign, these are primarily used to confirm the diagnosis of carpal tunnel syndrome (CTS) by eliciting paresthesia in the median nerve distribution. However, the sensitivity of these tests reported in the literature varies considerably — from 42% to 85% for Phalen’s test and from 38% to 100% for the Hoffmann-Tinel sign [Brüske, J., Bednarski, M., Grzelec, H., Zyluk, A. (2002). The usefulness of the Phalen test and the Hoffmann-Tinel sign in the diagnosis of carpal tunnel syndrome. Acta Orthop Belg, 68(2), 141-145]. This variability may contribute to the fact that the BCTQ-PL scales did not demonstrate sufficient discriminant power to significantly differentiate between patients with positive and negative results on these provocative tests. It is possible that in some cases, despite a negative test result, patients still experienced CTS symptoms that affected their functional status. We acknowledge that these inconsistencies highlight the complexity of CTS assessment and emphasize the need for a multidimensional approach, incorporating both objective and subjective tools, when evaluating patients with CTS.
The Discussion section has been supplemented: Lines 338 -346.
Issues with Structural Validity
Exploratory factor analysis (EFA) revealed that the Symptom Severity Scale (SSS) is composed of two factors rather than being unidimensional. Despite this, the authors continue to treat it as a single scale without proposing or discussing a potential subscale division. This raises concerns about the structural validity of the SSS, which should be explicitly addressed.
Thank you for this valuable comment. Our findings suggest the need for further investigation into the factorial structure of the BCTQ-PL. Future studies should apply confirmatory factor analysis (CFA) or Rasch modeling to verify the stability of this two-factor solution and determine whether a subscale division would be more appropriate. Importantly, such analyses should be conducted on larger and more diverse samples to enhance statistical power and generalizability. It is also worth noting that similar two-factor solutions for the SSS have been reported in previous validation studies of the BCTQ in other languages, which may reflect the inherent multidimensionality of symptom perception in CTS. However, since the original questionnaire was intended as a unidimensional measure and no consensus has yet been reached on a standardized subscale division, we have chosen to retain the original structure in this validation study, while indicating the need for further research.
The limitations section has been supplemented: Lines 384-389.
Errors in SF-36 Data Reporting
Anomalous values were found in the SF-36 results, particularly for the Mental Health and Social Functioning domains (e.g., means of 1.7 and 5.8), which are implausible. These inconsistencies suggest potential data reporting errors that should be corrected.
Thank you for pointing this out. We apologize for the oversight. We mistakenly included data representing the difference between the 3rd and 1st assessment in the SF-36 MCS. The correct data - SF-36 MCS scores from the 3rd assessment have now been included in Table 2.
Limited Scope of Validation Methods
The validation process relied solely on subjective patient-reported measures. No comparison with objective clinical evaluations (e.g., electromyography or clinical examination scores) was performed. Including such assessments would considerably strengthen the validation results.
Thank you for this valuable comment. We agree that the validation process primarily relied on subjective patient-reported outcome measures (PROMs), in accordance with the COSMIN guidelines for construct validity, which recommend comparison with instruments that measure similar, related but dissimilar, or unrelated constructs (with varying expected correlations). Objective clinical assessments, such as ENG or EMG, were not included in this study. However, we did include one objective measure - hand grip strength, which is reported in the lower sections of Tables 2 and 6. While we acknowledge that including additional objective clinical data would strengthen the validation process, the applied methodology is consistent with current psychometric standards for the validation of PROMs.
We have therefore addressed this limitation in the Limitations and Future Considerations section and highlighted the need for further studies involving objective clinical tools to enhance the validation of the BCTQ-PL. Lines 381-384.
Redundancy and Writing Style
The manuscript contains redundant statements, particularly regarding the reliability and validity of the BCTQ-PL, without adding new information. Improving the conciseness and focusing the discussion on critical methodological issues would enhance the quality of the manuscript.
Thank you for this valuable comment.
The discussion section has been supplemented:
Lines: 320-322
Lines 363-364
Lines 339-346
Potential Conflict of Interest Not Fully Addressed
While no conflicts of interest are declared, validating the questionnaire exclusively in patients treated with ESWT may introduce implicit bias. This aspect should be acknowledged and discussed.
Thank you for your valuable comment. We agree that validating the questionnaire solely in patients undergoing ESWT may introduce a potential source of bias. This limitation has been acknowledged and is discussed in the revised manuscript (see the Limitations and Future Considerations section Lines 375-381). We have also emphasized the need for future validation studies in broader and more heterogeneous patient populations receiving various types of treatment, to strengthen the generalizability and methodological robustness of the BCTQ-PL. At the same time, based on the obtained results, we conclude that the questionnaire may be considered a reliable tool for assessing outcomes in patients with CTS who have undergone physiotherapy. However, we refrain from generalizing its applicability to other forms of treatment, such as surgical intervention or corticosteroid injections, until further validation studies are conducted.
Reviewer 3 Report
Comments and Suggestions for Authors
I would like to thank the authors for their excellent research, and for presenting a well-prepared manuscript. The effort and attention to detail are evident, and this work makes a valuable contribution to the field.
General concept comments:
- While a considerable number of measures were taken as exclusion criteria, the inclusion criteria were not addressed correspondingly. For instance,, variables including “previous CTS treatment”, co-occurring diseases”, and limb domination with CTS” show a relatively biased distribution in the sample, that could have affected the findings. Acknowledging this issue in the Discussion section could enhance the transparency and credibility of the research.
- The construct validity is based on several well-thought a priori hypotheses. However, it is recommended to clearly link each hypothesis to its theoretical basis and empirical support.
- While reliability and validity tests were conducted thoroughly on the two sub-scales of the BCTQ-PL, results for the overall scale were not presented, limiting the comprehensive assessment of the tool's effectiveness.
- Although typically considered "Examination" tools, both the "Phalen’s Test" and the "Hoffmann-Tinel Test" have been described as standardized Patient-Reported Outcome Measures (PROMs) in the manuscript.
Specific comments:
- Consider replacing “p = 0.0000” with “p < 0.001” throughout the text.
- Consider replacing “,” with “.”, when reporting numbers in Tables 6 and 7.
- Page 11, sub-section 3.2.6.: Typing error for “BSTQ-PL”.
- If applicable, please also provide a “Scree Plot” for the construct validity section.
Author Response
Dear Reviewer,
We would like to express our gratitude to you for reviewing our article titled A Polish version of the Boston Carpal Tunnel Questionnaire (BCTQ-PL) for use among patients with Carpal Tunnel Syndrome undergoing physiotherapy: Translation, Cultural Adaptation, and Validation.
Please find detailed responses below and the corresponding corrections highlighted in the re-submitted files.
I would like to thank the authors for their excellent research, and for presenting a well-prepared manuscript. The effort and attention to detail are evident, and this work makes a valuable contribution to the field.
General concept comments:
- While a considerable number of measures were taken as exclusion criteria, the inclusion criteria were not addressed correspondingly. For instance,, variables including “previous CTS treatment”, co-occurring diseases”, and limb domination with CTS” show a relatively biased distribution in the sample, that could have affected the findings. Acknowledging this issue in the Discussion section could enhance the transparency and credibility of the research.
Thank you for this valuable comment. Limitations section has been supplemented: Lines: 377-381
Notably, the exclusion criteria applied, some of which were directly related to contraindications for ESWT, may have inadvertently introduced sample imbalances, particularly with respect to comorbidities and the distribution of CTS in dominant vs. non-dominant limbs. Future studies should aim to validate the BCTQ-PL in heterogeneous patient groups treated with different therapeutic approaches and recruited from various healthcare centers, in order to improve the generalizability of the findings.
- The construct validity is based on several well-thought a priori hypotheses. However, it is recommended to clearly link each hypothesis to its theoretical basis and empirical support.
Thank you for this valuable comment. Methods section has been supplemented: Lines: 167-184
(1) BCTQ-PL SSS will strongly correlate with QuickDASH (both tools assess symptom severity and functional impact of upper limb conditions, making a strong correlation theoretically expected); (2) BCTQ-PL FSS will strongly correlate with QuickDASH (FSS and QuickDASH both focus on functional limitations of the upper extremity, supporting a strong correlation); (3) BCTQ-PL SSS will strongly correlate with VAS (SSS includes pain and sensory disturbances, which are core elements measured by the VAS pain scale); (4) BCTQ-PL FSS will correlate moderately or weakly with VAS (FSS measures functional ability rather than pain intensity, suggesting a weaker theoretical link with VAS); (5) Correlations between BCTQ-PL and QuickDASH will be stronger than between BCTQ-PL and SF-36 (QuickDASH is disease-specific for upper limb function, whereas SF-36 is a generic quality-of-life tool, leading to expected stronger correlations with the BCTQ-PL); (6) Correlations between FSS and SF-36 will be stronger than between SSS and SF-36 ( FSS relates more directly to daily functioning, which aligns better with the functional domains of SF-36); (7) Correlations between FSS and SF-36 PCS will be stronger than between FSS and SF-36 MCS (FSS reflects physical disability, thus aligning more closely with the PCS of SF-36); (8) Correlations between SSS and SF-36 PCS will be stronger than between SSS and SF-36 MCS (symptom severity in CTS (e.g., pain, numbness) impacts physical health more directly than mental health); (9) Both BCTQ-PL scales will correlate moderately or weakly with hand grip strength (grip strength is an objective functional test, but often preserved in mild-to-moderate CTS, leading to limited correlation with subjective symptom reports); (10) Both BCTQ-PL scales will significantly differentiate patients with positive vs. negative results on Tinel-Hoffmann and Phalen’s tests (these provocative tests are diagnostic tools for CTS, so differences in SSS and FSS scores between positive and negative groups are theoretically expected).
- While reliability and validity tests were conducted thoroughly on the two sub-scales of the BCTQ-PL, results for the overall scale were not presented, limiting the comprehensive assessment of the tool's effectiveness.
Thank you for this valuable comment. We would like to clarify that we conducted the psychometric analyses in accordance with the original BCTQ authors’ conceptualization, who consider the scale as a two-factor construct and therefore performed separate analyses for the two subscales (SSS and FSS) without reporting an overall total score. This approach is also commonly applied in other language validation studies of the BCTQ, ensuring methodological consistency and comparability of results.
- Although typically considered "Examination" tools, both the "Phalen’s Test" and the "Hoffmann-Tinel Test" have been described as standardized Patient-Reported Outcome Measures (PROMs) in the manuscript.
Thank you for pointing this out.
Subsection 2.4 'Measurements' has been divided as follows:
2.4.1 – includes PROMs (2.4.1.1–2.4.1.4: BCTQ-PL, QuickDASH, SF-36, and VAS);
2.4.2 – includes provocative tests (2.4.2.1 and 2.4.2.2: Phalen’s Test and Hoffmann-Tinel sign);
2.4.3 – includes the muscle strength test (2.4.3.1: hand grip strength test using a dynamometer).
Specific comments:
- Consider replacing “p = 0.0000” with “p < 0.001” throughout the text.
Thank you for pointing this out. The changes have been made throughout the text.
- Consider replacing “,” with “.”, when reporting numbers in Tables 6 and 7.
Thank you for pointing this out. The changes have been made in Tables 6 and 7.
- Page 11, sub-section 3.2.6.: Typing error for “BSTQ-PL”.
Thank you for pointing this out. The typo has been corrected.
- If applicable, please also provide a “Scree Plot” for the construct validity section.
Thank you for pointing this out. We acknowledge the Reviewer’s suggestion and would like to clarify that, due to the number of variables and factors examined, we generated ten separate scree plots to illustrate the construct validity analyses. Including all of them in the manuscript might be excessive and could reduce clarity. However, we would be happy to provide the scree plots as supplementary material upon request, should the Reviewer find it useful.
Reviewer 4 Report
Comments and Suggestions for Authors
Dear Authors,
Thank you for submitting your manuscript entitled “A Polish version of the Boston Carpal Tunnel Questionnaire (BCTQ-PL) for use among patients with Carpal Tunnel Syndrome undergoing physiotherapy”.
Your work addresses an important and clinically meaningful need by providing a culturally adapted and validated Polish version of a widely used self-report tool for CTS. The methodology is clear, follows international standards, and the psychometric analysis is thorough.
That said, several points need clarification and refinement:
Major Comments
-
Abstract Clarity and Typo
-
Page 1: Final sentence includes a typo (“tthe condition of patients”). Please revise.
-
Suggest rewording the conclusion to:
“The BCTQ-PL is a valid and reliable tool for assessing CTS-related symptoms and functional status in Polish-speaking patients.”
-
-
Interpretation of Construct Validity
-
Pages 10–11, Table 7: Two of the 10 hypotheses were not confirmed (e.g., Phalen's test and SSS, p = 0.1620), yet the construct validity is described as “high.”
-
Recommendation: Revise to a more objective and cautious phrase such as
“Construct validity was moderate to high, with 80% of predefined hypotheses confirmed.”
-
-
Factor Analysis Interpretation
-
Page 11: The two-factor structure found in the SSS subscale requires clearer explanation. Interpretative labels (e.g., “daytime symptoms” vs. “nocturnal symptoms”) and clinical implications should be added.
-
If available, include supporting indices such as eigenvalues, variance explained, or KMO values.
-
-
Ethical Approval Section Placement
-
Page 7, 2.5.4: The ethics information appears abruptly before the Results section.
-
Suggest relocating this subsection to the end of “Materials and Methods” under the heading “Ethical Considerations” for consistency.
-
-
Figure and Table Referencing
-
Throughout pages 8–12, references to Tables 6–9 and Figures 1–2 are minimal and should be more explicitly integrated into the text (e.g., “as shown in Table 7”).
-
Minor Comments
-
Ensure all abbreviations (e.g., ICC, SSS, FSS) are consistently defined upon first use.
-
Table 2: SF-36 MCS scores appear abnormally low (e.g., 3.6). Please verify the data entry or clarify potential outliers.
Sincerely,
Author Response
Dear Reviewer,
We would like to express our gratitude to you for reviewing our article titled A Polish version of the Boston Carpal Tunnel Questionnaire (BCTQ-PL) for use among patients with Carpal Tunnel Syndrome undergoing physiotherapy: Translation, Cultural Adaptation, and Validation.
Please find detailed responses below and the corresponding corrections highlighted in the re-submitted files.
Comments and Suggestions for Authors
Dear Authors,
Thank you for submitting your manuscript entitled “A Polish version of the Boston Carpal Tunnel Questionnaire (BCTQ-PL) for use among patients with Carpal Tunnel Syndrome undergoing physiotherapy”.
Your work addresses an important and clinically meaningful need by providing a culturally adapted and validated Polish version of a widely used self-report tool for CTS. The methodology is clear, follows international standards, and the psychometric analysis is thorough.
That said, several points need clarification and refinement:
Major Comments
- Abstract Clarity and Typo
- Page 1: Final sentence includes a typo (“tthe condition of patients”). Please revise.
Thank you for pointing this out. The typo has been corrected.
- Suggest rewording the conclusion to: “The BCTQ-PL is a valid and reliable tool for assessing CTS-related symptoms and functional status in Polish-speaking patients.”
Thank you for your valuable suggestions. We have reworded the conclusion accordingly.
- Interpretation of Construct Validity
- Pages 10–11, Table 7: Two of the 10 hypotheses were not confirmed (e.g., Phalen's test and SSS, p = 0.1620), yet the construct validity is described as “high.”
- Recommendation: Revise to a more objective and cautious phrase such as
“Construct validity was moderate to high, with 80% of predefined hypotheses confirmed.”
Thank you for your valuable suggestions.
The Materials and Methods section has been corrected: Lines 185-186
According to the criteria proposed by Terwee et al. [26], construct validity of the BCTQ-PL is deemed acceptable when at least 75% of a priori hypotheses are confirmed in a sample of no fewer than 50 participants.
The Results section has been corrected: Lines 274-275
The construct validity of the BCTQ-PL received a positive rating according to the criteria of Terwee et al. [26], as 80% of the results were consistent with the predefined hypotheses.
The Discussion section has been corrected: Lines 338-339
In accordance with the guidelines of Terwee et al. [26], this result indicates acceptable construct validity of the BCTQ-PL.
- Factor Analysis Interpretation
- Page 11: The two-factor structure found in the SSS subscale requires clearer explanation. Interpretative labels (e.g., “daytime symptoms” vs. “nocturnal symptoms”) and clinical implications should be added.
- If available, include supporting indices such as eigenvalues, variance explained, or KMO values.
Thank you for your valuable suggestions.
The Materials and Methods section has been supplemented:
Lines: 190 – 193 (for each subscale, an eigenvalue greater than 1.0 was assumed, as well as an explained variance of more than 10%). The Kaiser–Meyer–Olkin measure of the sampling adequacy (KMO) was set at >0.70 to indicate adequate sampling, and the significance level of the Barlett Test of Sphericity was p < 0.001, indicating that the EFA could be used for data analysis.
The Results section has been supplemented:
Lines: 287-289 The KMO measure of sampling adequacy was 0.792 for the SSS and 0.909 for the FSS, indicating a good fit for factor analysis. Bartlett’s test of sphericity was significant for both the SSS and FSS (p < 0.001), suggesting that the correlation matrices differ significantly from the identity matrix and justifying the use of the EFA.
Supporting indices, including eigenvalues and the variance explained, are provided in Table 8.
- Ethical Approval Section Placement
- Page 7, 2.5.4: The ethics information appears abruptly before the Results section.
- Suggest relocating this subsection to the end of “Materials and Methods” under the heading “Ethical Considerations” for consistency.
Thank you for pointing this out. The ethics information has been placed under the heading 2.6 Ethical Considerations. This section is located at the end of “Materials and Methods”, Lines 203-205.
- Figure and Table Referencing
- Throughout pages 8–12, references to Tables 6–9 and Figures 1–2 are minimal and should be more explicitly integrated into the text (e.g., “as shown in Table 7”).
Thank you for pointing this out. References to Tables 6–9 (and also for Table 4) and Figures 1–2 have been revised accordingly:
- Figure 1 presents a flow chart depicting the recruitment process and subsequent testing stages of patients.
- Figure 2 presents the six steps of the linguistic and cultural adaptation of the BCTQ-PL.
- As shown in Tables 6 and 7, a-priori hypotheses No. 1 through 8 were confirmed, while hypotheses No. 9 and 10 were rejected.
- The results of the EFA for the BCTQ-PL are presented in Table 8.
- As shown in Table 9, the mean decrease in the severity of symptoms was 1.04 points, and the functional status improved by 0.77 points. …..
- As shown in Table 4, the mean decrease in the severity of symptoms was 1.04 points, and the functional status improved by 0.77 points. …
Minor Comments
- Ensure all abbreviations (e.g., ICC, SSS, FSS) are consistently defined upon first use.
Thank you for pointing this out.
The expanded forms of the abbreviations have been provided in:
the Abstract: lines 24–27
the Materials and Methods section, lines: 66, 79, 139-140
- Table 2: SF-36 MCS scores appear abnormally low (e.g., 3.6). Please verify the data entry or clarify potential outliers.
Thank you for pointing this out. We apologize for the oversight, we mistakenly included data representing the difference between the 3rd and 1st assessment in the SF-36 MCS. The correct data - SF-36 MCS scores from the 3rd assessment have now been included in Table 2.
Round 2
Reviewer 2 Report
Comments and Suggestions for Authors
Thank you